# Chronic Administration of Non-Constitutive Proteasome Inhibitor Modulates Long-Term Potentiation and Glutamate Signaling-Related Gene Expression in Murine Hippocampus

**DOI:** 10.3390/ijms24098172

**Published:** 2023-05-03

**Authors:** Alexander Maltsev, Sergei Funikov, Alexander Rezvykh, Ekaterina Teterina, Vladimir Nebogatikov, Alexander Burov, Natalia Bal, Aleksey Ustyugov, Vadim Karpov, Alexey Morozov

**Affiliations:** 1Institute of Higher Nervous Activity and Neurophysiology, Russian Academy of Sciences, Butlerova 5A, 117485 Moscow, Russia; alexmaltsev2504@gmail.com (A.M.); bal_nv@mail.ru (N.B.); 2Engelhardt Institute of Molecular Biology, Russian Academy of Sciences, Vavilov Street 32, 119991 Moscow, Russia; sergeifunikov@mail.ru (S.F.); aprezvykh@yandex.ru (A.R.); alexanderburov1998@gmail.com (A.B.); karpovvl2008@gmail.com (V.K.); 3Institute of Physiologically Active Compounds at Federal Research Center of Problems of Chemical Physics and Medicinal Chemistry, Russian Academy of Sciences, Severny Proezd, 1, 142432 Chernogolovka, Russia; ekaterina_nia14@mail.ru (E.T.); vnebogatikov@gmail.com (V.N.); alexey.ipac@gmail.com (A.U.)

**Keywords:** proteasome, non-constitutive proteasomes, proteasome inhibitor, ONX-0914, synaptic plasticity

## Abstract

Proteasomes degrade most intracellular proteins. Several different forms of proteasomes are known. Little is known about the role of specific proteasome forms in the central nervous system (CNS). Inhibitors targeting different proteasome forms are used in clinical practice and were shown to modulate long-term potentiation (LTP) in hippocampal slices of untreated animals. Here, to address the role of non-constitutive proteasomes in hippocampal synaptic plasticity and reveal the consequences of their continuous inhibition, we studied the effect of chronic administration of the non-constitutive proteasome inhibitor ONX-0914 on the LTP induced by two different protocols: tetanic stimulation and theta-burst stimulation (TBS). Both the tetanus- and TBS-evoked potentiation contribute to the different forms of hippocampal-dependent memory and learning. Field-excitatory postsynaptic potentials (fEPSPs) in hippocampal slices from control animals and animals treated with DMSO or ONX-0914 were compared. LTP induced by the TBS was not affected by ONX-0914 administration; however, chronic injections of ONX-0914 led to a decrease in fEPSP slopes after tetanic stimulation. The observed effects correlated with differential expression of genes involved in synaptic plasticity, glutaminergic synapse, and synaptic signaling. Obtained results indicate that non-constitutive proteasomes are likely involved in the tetanus-evoked LTP, but not the LTP occurring after TBS, supporting the relevance and complexity of the role of specific proteasomes in synaptic plasticity, memory, and learning.

## 1. Introduction

Proteasomes degrade most intracellular proteins. The 20S proteasome represents a protein complex assembled from multiple alpha and beta subunits. Three of the beta subunits: β1, β2, and β5, are catalytically active and cleave the peptide bonds after acidic, basic, and hydrophobic amino acids, correspondingly [1]. During proteasome assembly, constitutive beta subunits might be replaced by so-called «immune» analogs (β1i, β2i, and β5i), giving rise to immune or intermediate (if not all of the constitutive beta subunits were replaced) proteasomes [2]. These non-constitutive proteasomes were shown to play important roles in immune reactions as their capacity to generate MHC-I-compatible peptides is higher. Accordingly, their levels are elevated in the immune cells or can be upregulated in different somatic cells under stress or inflammatory conditions [3]. At the same time, accumulating evidence suggests an important role of non-constitutive proteasomes in various tissues in the normal state, highlighting their specific metabolic implications [2].

Non-constitutive proteasomes were revealed in different cells of the central nervous system (CNS), including astrocytes, microglia, and neurons, though in much lower quantities [4,5,6]. However, the role of these proteasomes in the brain is poorly understood. Several papers indicate that activation of non-constitutive proteasomes within the cells of the CNS is associated with inflammation and observed in neurodegenerative diseases [4,5]. Beyond that, proteasomes were demonstrated to play an important role in synaptic plasticity and memory formation [7,8,9,10]. Inhibition of the entire pool of proteasomes in the hippocampal slices with broad-specificity inhibitors (targeting both constitutive and non-constitutive proteasomes) was shown to modulate long-term potentiation (LTP), which is the most studied form of synaptic plasticity, associated with learning and memory [7,8,9]. At the same time, the molecular mechanism underlying this phenomenon is still mostly unclear, and the role of non-constitutive proteasomes in this form of plasticity was not addressed. Recently, to test the involvement of non-constitutive proteasomes in the LTP induction, we incubated the hippocampal slices obtained from untreated healthy mice with the β5i-specific inhibitor—ONX-0914 [10]. Suppressed LTP kinetics in treated samples was observed, indicating participation of the β5i-containing non-constitutive proteasomes in synaptic plasticity [10]. In the near future, non-constitutive proteasome inhibitors might be used as therapeutic opportunities in the treatment of different diseases; however, the effects of their prolonged administration on the synaptic plasticity are unknown. Therefore, we sought to study the LTP after different stimulation paradigms in hippocampal slices of animals that were receiving the ONX-0914 for a considerable amount of time. To gain insight into the molecular mechanisms, we compared patterns of gene expression in the hippocampi of control and ONX-0914-treated mice.

## 2. Results

### 2.1. Chronic Administration of ONX-0914 Modulates LTP in the Hippocampus Following Tetanic Stimulation, but Not Short Theta-Burst Stimulation

To characterize LTP kinetics, we have recorded field-excitatory postsynaptic potentials (fEPSPs) in hippocampal slices of control CD1 mice and animals treated for two months with ONX-0914 or DMSO (Figure 1a). The fEPSP slope is the tangent of the tilt angle (tg α) for the descending part of the postsynaptic response, corresponding in meaning to the rate of change in postsynaptic potentials (Appendix A). The recording of fEPSP slopes revealed that LTP kinetics induced by ‘weak’ TBS was identical in hippocampal slices in all studied groups, indicating that non-constitutive proteasomes are likely not involved in the TBS-induced synaptic plasticity (Figure 1b,c). Thus, 60 min after TBS induction, fEPSP slopes were: 141.2 ± 12.0% in the control group, 143.6 ± 8.2% in the ONX-0914-treated group, and 136.1 ± 9.8% in the DMSO-treated group (for all, n = 6, *p* ˃ 0.05, Mann–Whitney U test). Furthermore, 1 h of incubation of hippocampal slices from the three tested groups in the presence of 100 nM of ONX-0914 did not reveal any differences in the fEPSP dynamics during LTP following TBS (Figure 1b,c).

Next, we estimated the effect of prolonged ONX-0914 administration on the LTP occurring after ‘strong’ tetanic stimulation. In our previous experiments, we observed suppressed LTP kinetics in the C57BL/6 mice hippocampal slices after 1 h of incubation with 100–250 nM of ONX-0914 [10]. Here, we revealed that chronic administration of ONX-0914 to CD1 mice led to a significant decrease in fEPSP slopes after tetanic stimulation. Two hours after LTP induction, the fEPSP slope in samples from ONX-0914-treated animals was 141.0 ± 16.2%, while in control non-injected or DMSO-treated CD1 mice, the slopes were 195.4 ± 6.8% and 188.9 ± 13.4%, respectively (for all, n = 6, *p* ˂ 0.05, Mann–Whitney U test, vs. ONX-injected group) (Figure 1b,c). The incubation of slices with 100 nM of ONX-0914 leveled the differences in all groups, showing the impairment of synaptic plasticity in control non-injected (fEPSP slope 126.9 ± 9.0%) or DMSO-injected CD1 mice (fEPSP slope 127.5 ± 10.1%) during transient non-constitutive proteasome blockade (for all, n = 6, *p* ˂ 0.05, Mann–Whitney U test, vs. untreated slices) (Figure 1b,c).

### 2.2. Chronic Administration of ONX-0914 Results in Altered Expression of Glutamate Signaling-Related Genes

To address the molecular mechanism behind LTP modulation, we performed RNA-seq of hippocampi obtained from control (untreated), ONX-0914-, and DMSO-treated animals. The analysis of RNA-seq data showed a significant difference in gene expression profiles in samples from ONX-0914-treated animals compared with samples from both DMSO-treated and control mice, which can be seen in multidimensional scaling (Figure 2a). The performed analysis revealed 304 differentially expressed genes (DEG, FDR < 0.05) after ONX-0914 administration, while only 12 DEG were observed following DMSO treatment (Figure 2b). A set of DEG has been confirmed by quantitative PCR (data not shown). For further characterization, we kept 271 protein-coding genes whose expression was not affected by DMSO without ONX-0914 treatment (Figure 2b). Among the DEG in hippocampi of ONX-914-treated mice, 110 and 161 demonstrated downregulation and upregulation, respectively (Figure 2b,c). Notably, all the observed DEG showed expression level changes not less than 2-fold (Log2FC = 1 or −1), indicating that chronic administration of ONX-0914 results in pronounced gene expression alterations (Figure 2c).

Among the genes that exhibited the most significant expression changes, we observed a drastic decrease in expression levels of genes encoding a vesicular glutamate transporter, VGLUT3 (*Slc17a8*, Log2FC = −1.79), glutamate receptors including NMDA receptor (*Grin2b*, Log2FC = −1.05) and AMPA (*Gria1*, Log2FC = −1.57, and *Gria2*, Log2FC = −1.01), as well as glutamate metabotropic receptor (*Grm4*, Log2FC = −1.5) (FDR for all genes < 0.05). We also observed significant downregulation of mitogen-activated protein kinase 1 (*Map2k1*, Log2FC = −1.15), G protein-coupled receptor for serotonin (*Htr1a*, Log2FC = −1.24), cholinergic receptor nicotinic alpha 2 subunit (*Chrna2*, Log2FC = −2.34), and the downregulation of alpha-synuclein (*Snca*, Log2FC = −1.47) (FDR for all genes < 0.05). To annotate DEG in terms of molecular pathways in which gene products are involved, we performed the Gene Set Enrichment Analysis (GSEA). As a result, we observed that the most enriched categories in terms of Gene Ontology (GO) biological processes were: “regulation of postsynaptic membrane potential” (FDR = 8.44 × 10^−6^), “regulation of synapse structure or activity” (FDR = 4.09 × 10^−6^), and “synaptic signaling” (FDR = 7.08 × 10^−9^). It is known that the LTP is induced by stimulation of glutamatergic synapses [11]. Notably, many DEG were enriched in “glutamatergic synapse” (FDR = 2.99 × 10^−7^) and “glutamate receptor activity” (FDR = 8.39 × 10^−7^) in terms of GO cellular component and GO molecular function, respectively (Figure 2d). The most enriched KEGG pathways by DEG were “long-term potentiation” (FDR = 0.003) and “glutamatergic synapse” (FDR = 0.0003). We observed that 8 of 9 significantly enriched categories in both GO and KEGG terms exhibited a decreasing trend of their expression (more than 50% of genes that fall into these categories show Log2FC < 0).

## 3. Discussion

Long-term potentiation (LTP) is considered one of the most important tissues’ correlates for learning and memory. It occurs in the hippocampal CA1 dendrites after high-frequency stimulation of presynaptic Shaffer’s collaterals from the CA3 layer [12,13]. The LTP is widely used as a quantitative parameter in modeling memory impairments and screening for potential protectors against these disorders. Thus, the prevention of damaging factor-induced LTP inhibition, in turn, partially compensates for cognitive deficits in various experimental models of cognitive impairments, including behavioral changes caused by the neurodegenerative diseases [7,8,12]. One of the key mechanisms for the LTP maintenance at the level of individual neurons is a change in the activity of cellular proteins, including ion channels, receptors, and functional enzymes, which provides the increased excitability of postsynaptic endings for a considerable time after the impact on the afferent pathways [7,13]. In several publications using broad-specificity proteasome inhibitors, it was shown that proteasomes participate in the induction of LTP [9]. However, the role of specific forms of proteasomes was not investigated. Previously, we demonstrated that the non-constitutive proteasome inhibitor ONX-0914 suppressed the LTP induction in hippocampal slices obtained from healthy mice [10]. To gain more information on the role of non-constitutive proteasomes in the CNS, and since non-constitutive proteasome inhibitors represent promising therapeutic agents [14] being evaluated in clinical trials as possible drugs against severe human pathologies, we sought to study the effect of chronic administration of the ONX-0914 on the LTP. To this end, we used two different stimulation paradigms: ‘weak’ theta-burst stimulation (TBS) or ‘strong’ tetanic (tetanus) stimulation of CA3-CA1 inputs. Phillips et al. demonstrated that molecular mechanisms for LTP arising from the TBS and tetanic stimulation are distinct, which can be useful to optimize the hippocampal circuit’s functioning and behavioral performance [15]. Following TBS, we observed no changes in the LTP kinetics in hippocampal slices obtained from control and inhibitor-treated CD1 animals. Interestingly, a previous report indicated strengthening of the LTP after TBS following incubation with a broad-specificity proteasome inhibitor [8]. No effect of ONX-0914 in our experiments suggests that non-constitutive proteasomes might not be involved in the TBS-induced synaptic plasticity and that different forms of proteasomes are likely differently implicated in the TBS-evoked plasticity in the CA3-CA1 synapses. On the other hand, the LTP was significantly suppressed in the hippocampi of ONX-0914-treated but not control mice following tetanic stimulation, which was in line with our previous findings. This effect might be associated with several mechanisms. After experiments with broad-specificity proteasome inhibitors, Fonseca and coauthors proposed that LTP modulation is associated with the disturbed balance between synthesis and degradation of proteins that affect plasticity [7]. On the other hand, non-constitutive proteasomes were shown to modulate the expression of various genes [16], including those involved in regulation of the inflammation [17], which in turn can influence the LTP [18]. Analysis of hippocampal transcriptomes of ONX-0914-treated animals revealed significant expression level changes and repression of genes involved in synaptic signaling. Of note, genes encoding glutamate transporter VGLUT3 and glutamate receptors NMDA and AMPA were significantly downregulated. The importance of these genes relies on the fact that the LTP is induced by the stimulation of glutamatergic synapses composed of NMDA and AMPA receptors. Changes in the glutamate release, AMPA receptor properties, and the number were proposed as potential mechanisms of LTP expression [19]. Moreover, G protein-coupled receptors for serotonin were demonstrated to influence the synaptic plasticity via modulation of NMDA receptor activity [20]. Furthermore, it has been shown that conditional expression of a mutant form of MEK1 (encoded by *Map2k1*) in the postnatal murine forebrain inhibited ERK activation and caused selective deficits in the translation-dependent, transcription-independent phase of hippocampal LTP [21]. Therefore, our data indicate that chronic administration of ONX-0914 can affect LTP by modulation of expression of several key genes, involved in the regulation of glutamatergic synapse activity and downstream signaling pathways. Interestingly, upon 1 h of incubation of slices obtained from control animals with the inhibitor, we also observed decreased LTP after tetanic stimulation. Since the incubation time with the ONX-0914 was rather short, one cannot exclude that activation of mechanisms other than gene expression modulation led to the decreased LTP. Here, recently discovered neuronal membrane-specific proteasomes could be mentioned [22,23]. These proteasomes are integrated into the neuronal membrane and are required for learning-induced behavioral plasticity. Since neuronal membrane-specific proteasomes are exposed to the extracellular environment, the observed effect of LTP modulation after short-term incubation of hippocampal slices from untreated animals with the ONX-0914 might be partially associated with changes in the activity of these complexes. Thus, it would be interesting to address the influence of the ONX-0914 on membrane-specific proteasomes in more detail.

## 4. Materials and Methods

### 4.1. Animals

Male aged-matched outbred CD1 mice were used in the study. Mice were maintained under SPF conditions at the Center for Collective Use of the Institute of Physiologically Active Compounds, Russian Academy of Sciences (#FFSN-2021-0005). Animals were housed in groups of five per cage in a standard environment (12 h light/dark cycle, 18–26 °C room temperature, and 30–70% relative humidity) with food and water ad libitum. Work with animals was carried out in accordance with the “Rules of laboratory practice in the Russian Federation”, dated 01.04.2016, No. 199n.

### 4.2. Administration of ONX-0914

Mice were either receiving 10 mg/kg of ONX-0914 (Apexbio, Houston, TX, USA) [24] (n = 6) or an equal volume of DMSO (n = 6) intraperitoneally twice a week during a two-month period. An additional group of CD1 animals (n = 6) received no treatment.

### 4.3. Preparation and Treatment of Hippocampal Slices

Animals were anesthetized by sevoflurane and decapitated. Brains were quickly submerged in ice-cold dissection solution (124 mM NaCl, 3 mM KCl, 1.25 mM NaH_2_PO_4_, 26 mM NaHCO_3_, 0.5 mM CaCl_2_, 7 mM MgCl_2_, and 10 mM D-glucose, pH equilibrated with 95% O_2_—5% CO_2_). Parasagittal hippocampal slices (350 µm thickness) were prepared using a vibratome (Leica VT1000S, Wetzlar, Germany) and immediately transferred to a recording solution (ACSF, composition as above, except the CaCl_2_ and MgCl_2_ concentrations were adjusted to 2.5 and 1.3 mM, respectively). Slices were heated to 34 °C in a water bath for 40 min and then kept at room temperature. Additionally, hippocampal slices were incubated for 1 h with (ONX-treated groups) or without (control groups) ONX-0914 (100 nM in the ASCF solution).

### 4.4. Electrophysiology

Hippocampal slices were perfused by a continuous ACSF flow (approximately 4 mL/min) at 32–33 °C. Electrophysiological recordings were carried out using a SliceMaster system (Scientifica, Uckfield, UK). Field-excitatory postsynaptic potentials (fEPSPs) were recorded from *Stratum radiatum* in the CA1 area using glass microelectrodes (1–2 MΩ) filled with the chamber solution. Baseline synaptic responses were evoked by paired-pulse stimulation of the Schaffer collaterals with a 50 ms interval at 0.033 Hz, with a bipolar electrode. The test stimulation intensity was adjusted to evoke fEPSPs with an amplitude of 50% of the maximal and was kept constant throughout the experiment. TBS contained 4 trains with 4 stimuli per each train, spaced 30 s apart, as in [13], while ‘strong’ tetanic stimulation (tetanus) was performed with four 100 Hz trains spaced 5 min apart, as in [10]. The data were recorded and analyzed by Spike2 and SigmaPlot 11.0 (Systat Software Inc., San Jose, CA, USA). For statistical analysis, the first 2 min after TBS or tetanus induction, intermediate 58–60 min, and 118–120 min for both stimulation paradigms were used. The average values of the fEPSP slopes during 20 min (1 measure every 30 s, total 40 sweeps) before the induction of LTP were taken as 100% (control) (Appendix A). For baseline responses, the appropriate fEPSP slopes during the test stimulation throughout the experiment were evaluated. The electrophysiological experiments were carried out using equipment of the Research Resource Center # 40606 of IHNA and NPh RAS ‘Functional Brain Mapping’.

### 4.5. RNA Extraction and RNA-Seq

RNA extraction, libraries’ preparation for RNA-seq, and differential gene expression analysis were performed as described in [25]. Briefly, total RNA was extracted using the Extract RNA reagent (Evrogen, Moscow, Russia). The RNA Integrity Number (RIN) of all RNA samples was not less than 8, as measured by the Agilent BioAnalyzer 2100 using an RNA 600 nano kit (Agilent Technologies, Santa Clara, CA, USA). Libraries for RNA-seq were prepared using the NEBNext^®^ Ultra™ II Directional RNA Library Prep Kit for Illumina (New England Biolabs, Ipswich, MA, USA), according to the manufacturer’s guidelines. One hundred bp single-end sequencing was conducted on an Illumina NextSeq 2000 platform (Illumina, San Diego, CA, USA). Differential expression calculations were performed in R language with the edgeR package using a quasi-likelihood, negative binomial, generalized log-linear model (glmQLFtest function) [26]. Successive multiple corrections of *p*-values were calculated using the Benjamini–Hochberg method. Gene Set Enrichment Analysis (GSEA) was performed using ShinyGO [27].

## 5. Conclusions

The obtained results indicated that the chronic intraperitoneal administration of a non-constitutive proteasome inhibitor affected the tetanus-evoked LTP, but not the LTP occurring after the ‘weak’ theta-burst stimulation. Though putative unspecific side effects of the inhibitor could not be entirely ruled out, our data indicated possible engagement of non-constitutive proteasomes and highlighted their role in the modulation of neuronal plasticity in the central nervous system. Moreover, these effects should be taken into consideration when broad-specificity and form-specific proteasome inhibitors are administered.

## Figures and Tables

**Figure 1 ijms-24-08172-f001:**
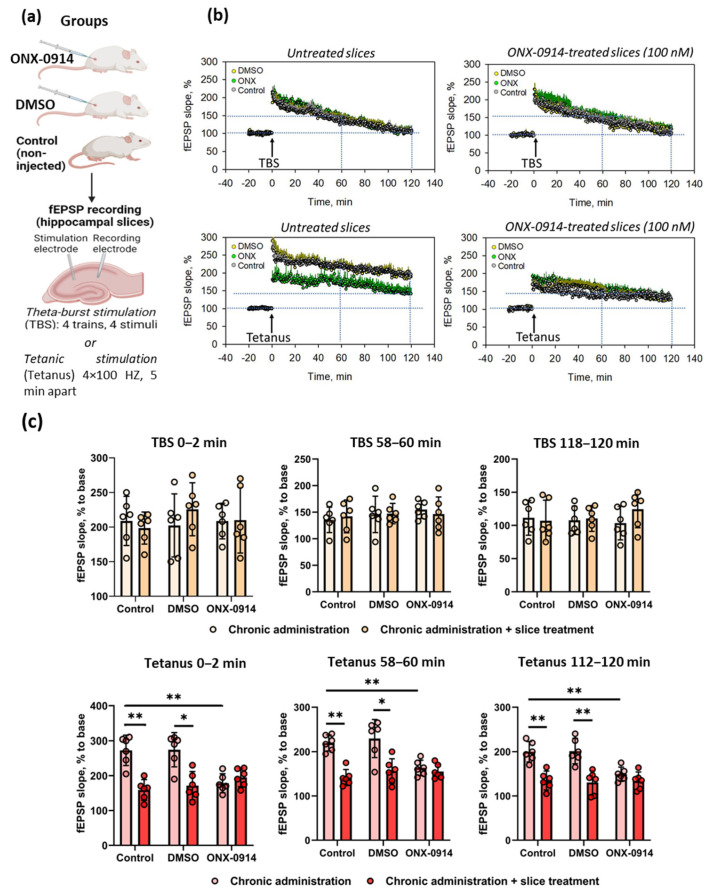
Long-term pharmacological inhibition of non-constitutive proteasomes modulates plasticity in the hippocampus after tetanic stimulation, but not after short theta-burst stimulation. (**a**) Graphic representation of the experimental design. (**b**) Left panel: Long-term potentiation (LTP) induced by theta-burst stimulation (TBS) or tetanic stimulation of hippocampal slices obtained from control animals and mice chronically (2 months) treated with DMSO or 10 mg/kg of ONX-0914. Right panel: LTP after TBS or tetanic stimulation of hippocampal slices obtained from control animals and mice chronically (2 months) treated with DMSO or 10 mg/kg of ONX-0914 following additional treatment with 100 nM of ONX-0914 for 1 h. Dynamic changes of fEPSP slopes are shown in (**c**). fEPSP slope at the early (0–2 min), middle (58–60 min), or late (118–120 min) stage of LTP. Quantitative assessment of data (**b**). Statistical significance was tested using the Mann–Whitney U test, * *p* ˂ 0.05, ** *p* ˂ 0.05.

**Figure 2 ijms-24-08172-f002:**
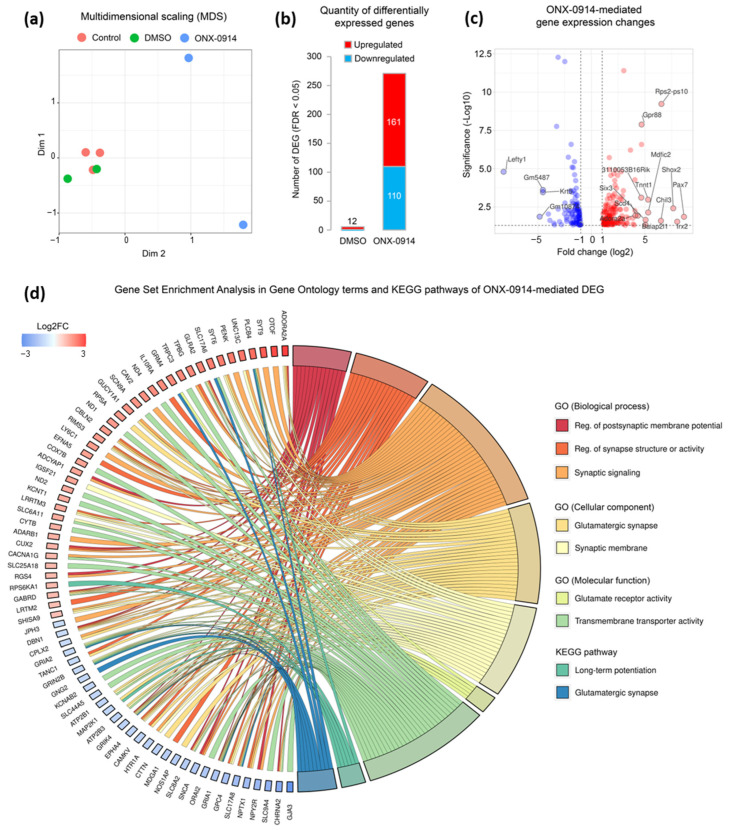
Chronic administration of ONX-0914 influences the gene expression profile in mouse hippocampus. (**a**) Multidimensional scaling (MDS) plot of pairwise distances calculated using root-mean-square of Log2FC values for all studied samples, including those from DMSO- and ONX-0914-treated mice. (**b**) Number of differentially expressed genes (DEG) after DMSO and ONX-0914 treatment (FDR < 0.05). (**c**) Volcano plot demonstrating dynamics of gene expression changes (upregulation and downregulation) after chronic administration of ONX-0914 in comparison to non-treated mice. (**d**) Gene Set Enrichment Analysis (GSEA) in Gene Ontology terms and KEGG pathways. Gene Ontology terms include three categories—biological function (molecular events and pathways in which gene products are involved), cellular component (the parts of a cell or its extracellular environment in which the gene product operates), and molecular function (the specific activity of a gene product at the molecular level). KEGG pathways represent a systematic analysis of gene functions, linking genomic information with higher-order functional information. The circular plot depicts DEG linked to enriched functional categories. Only terms and pathways with an enrichment FDR not less than 0.01 are shown. Genes are shown with their expression level changes (Log2FC, FDR < 0.05).

## Data Availability

RNA-seq data reported in this study are deposited in NCBI GEO under the number GSE229000.

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
