# Peer review of "Chronic Administration of Non-Constitutive Proteasome Inhibitor Modulates Long-Term Potentiation and Glutamate Signaling-Related Gene Expression in Murine Hippocampus"

_ijms, 2023, doi:10.3390/ijms24098172_

Round 1

Reviewer 1 Report

  1. Reorganize sentences for clarity and coherence: Rearrange the sentences to create a clearer and more coherent flow of information. Begin with a brief introduction to proteasomes, their types, and their role in protein degradation. Then, provide information about non-constitutive proteasomes, their involvement in immune reactions, and their presence in the central nervous system (CNS).
  2. Define technical terms and abbreviations: Ensure that all technical terms and abbreviations are defined when first introduced. For example, explain what long-term potentiation (LTP) is and why it is important for synaptic plasticity.
  3. Improve sentence structure and grammar: Check the text for grammatical errors and awkward phrasing. Revise sentences to improve readability and comprehension.
  4. Provide context for the current study: Clearly explain the knowledge gap and the significance of investigating the role of non-constitutive proteasomes in synaptic plasticity. Briefly mention the methods used in the study, such as chronic administration of ONX-0914, and the goals of the study.
  5. Use more specific language to describe the results of the gene expression analysis, for example by indicating which genes showed the most significant changes in expression levels and what functional categories they belong to.

Author Response

Dear Reviewer #1,

Thank You very much for careful reading of the manuscript, valuable comments and suggestions. We have modified the text accordingly. Please find point-by-point responses below

1) Reorganize sentences for clarity and coherence: Rearrange the sentences to create a clearer and more coherent flow of information. Begin with a brief introduction to proteasomes, their types, and their role in protein degradation. Then, provide information about non-constitutive proteasomes, their involvement in immune reactions, and their presence in the central nervous system (CNS).

Thank You for the comment. We modified the text in order to improve readability of the manuscript and carefully checked it for grammatical errors with the native speaker. Many sentences were rearranged and parts of the text were replaced. Introduction was significantly modified.

2) Define technical terms and abbreviations: Ensure that all technical terms and abbreviations are defined when first introduced. For example, explain what long-term potentiation (LTP) is and why it is important for synaptic plasticity.

Corrected. Technical terms and abbreviations were defined. In the Discussion section we clarified: «LTP is widely used as a quantitative parameter in modeling memory impairments and screening for potential protectors against these disorders. Thus, the prevention of damaging factor-induced LTP inhibition, in turn, partially compensates for cognitive deficits in various experimental models of cognitive impairments, including behavioral changes caused by the manifestation of neurodegenerative diseases [7,8,11]. One of the key me-chanisms for the LTP maintenance at the level of individual neurons is a change in the activity of cellular proteins, including ion channels, receptors, and functional enzymes, which provides the increased excitability of postsynaptic endings for a considerable time after the impact on the afferent pathways [7,12]».

Moreover, we introduced the information regarding the fEPSP slope into the Results and Materials and Methods sections. We have also added the Suppl. Fig. 1, to clearly demonstrate a slope of fEPSP. “The fEPSP slope is the tangent of the tilt angle (tg α) for the descending part of the postsynaptic response, corresponding in meaning to the rate of change in postsynaptic potentials. The 100% slope corresponds to the tangent of the tilt angle for the postsynaptic responses before the tetanus or TBS induction in the same slice, whereby an increase in the slope above 100% corresponds to a potentiation caused by one or another electrical stimulation protocol”.

Suppl. Fig. 1. (Please find the Figure in the attachment) The fEPSP slope estimation in the CA3-CA1 synaptic inputs of hippocampal slices. There are representative recordings for the fEPSP responses in the same slice for: 1 – control recording, before Tetanic induction (black curve), 2 – immediately after Tetanic induction (green curve), and 3 – the 2h after Tetanic induction (dark yellow curve). The fEPSP slope is the tangent of the tilt angle (α1, α2, and α3 for the 1st, 2d, and 3d state of slice throughout experiment, respectively) for the descending part of the postsynaptic response. The average values of the slopes during 20 min (1 measure every 30 s, total 40 sweeps) before the induction of LTP were taken as 100% (control).

3) Improve sentence structure and grammar: Check the text for grammatical errors and awkward phrasing. Revise sentences to improve readability and comprehension.

Done. The manuscript was carefully checked for grammatical and other errors by the native speaker.

4) Provide context for the current study: Clearly explain the knowledge gap and the significance of investigating the role of non-constitutive proteasomes in synaptic plasticity. Briefly mention the methods used in the study, such as chronic administration of ONX-0914, and the goals of the study.

We modified the abstract: «Here to address the role of non-constitutive proteasomes in hippocampal synaptic plasticity and reveal the consequences of their continuous inhibition we studied the effect of chronic administration of the non-constitutive proteasome inhibitor ONX-0914 on the LTP induced by two different protocols: tetanic stimulation and theta-burst stimulation (TBS). Both the tetanus- and TBS-evoked potentiation contribute to the different forms of hyppocampal-dependent memory and learning. ».

In the introduction we highlighted the rationale of the study. “Suppressed LTP kinetics in treated samples was observed, indicating participation of the β5i-containing non-constitutive proteasomes in synaptic plasticity [10]. In the near future non-constitutive proteasome inhibitors might be used as therapeutic opportunities in treatment of different diseases; however effects of their prolonged administration on the synaptic plasticity are unknown. Therefore, we sought to study the LTP after different stimulation paradigms in hippocampal slices of animals that were receiving the ONX-0914 for a considerable amount of time. To gain insight into the molecular mechanism we compared patterns of gene expression in the hippocampi of control and ONX-0914 treated mice”.

The Results and Discussion sections were accordingly modified.

5) Use more specific language to describe the results of the gene expression analysis, for example by indicating which genes showed the most significant changes in expression levels and what functional categories they belong to

Done. We rewrote the part of the manuscript with the transcriptome analysis. We also significantly expanded the figure 2 caption and introduced additional details.

Reviewer 2 Report

Line 73: “…fEPSP slopes were 141.2 ± 12.0%, 143.6 ± 8.2%, 136.1 ± 9.8%, for control non-injected, ONX0914-, and DMSO-treated groups…”

The readers of the article will not only be electrophysiologists, so it is necessary to explain what a slope is and what was taken as a 100% slope?

Line 83:  188.9 ± 13.4%

The work is devoted to the traditionally important issue of the specificity of the inhibitor effect on signaling in a complex system, which is a brain slice. In particular, the ability ONX-0914, the inhibitor of the 5Bi subunit of non-constitutive proteasomes, to affect LTP induced by tetanic or theta-burst stimulation. In addition to electrophysiological measurements, RNA expression analysis was performed and differential expression parameters of more than 300 mRNA were calculated.is to modulate long-term potentiation (LTP) in hippocampal slices

The topic is relevant in the study of mechanisms of non-constitutive proteasomes functioning in neuronal plasticity.  

The data obtained show for the first time that inhibition of the β5i subunit of non-constitutive proteasomes affects the glutamatergic transmission by reducing the expression of the NMDA- and AMPA-receptors mRNA.

The readers of the article will not only be electrophysiologists, so it is necessary to explain what a slope is and what was taken as a 100% slope?

The conclusions are consistent with the evidence.

The references are appropriate.

The manuscript does not contain tables. Figure 2a needs in a more detailed description.

Author Response

Dear Reviewer #2,

Thank You very much for careful reading of the manuscript, valuable comments and suggestions. We have modified the text accordingly. Please find point-by-point responses below

  • Line 73: “…fEPSP slopes were 141.2 ± 12.0%, 143.6 ± 8.2%, 136.1 ± 9.8%, for control non-injected, ONX0914-, and DMSO-treated groups…”

Corrected. The sentence was modified.

  • The readers of the article will not only be electrophysiologists, so it is necessary to explain what a slope is and what was taken as a 100% slope?

We added the information regarding the fEPSP slope into the Results and M&M sections. In addition we added Suppl. Fig. 1, to clearly demonstrate a slope of fEPSP. “The fEPSP slope is the tangent of the tilt angle (tg α) for the descending part of the postsynaptic response, corresponding in meaning to the rate of change in postsynaptic potentials. The 100% slope corresponds to the tangent of the tilt angle for the postsynaptic responses before the tetanus or TBS induction in the same slice, whereby an increase in the slope above 100% corresponds to a potentiation caused by one or another electrical stimulation protocol”.

Suppl. Fig. 1. (Please find the Figure in the attachment) The fEPSP slope estimation in the CA3-CA1 synaptic inputs of hippocampal slices. There are representative recordings for the fEPSP responses in the same slice for: 1 – control recording, before Tetanic induction (black curve), 2 – immediately after Tetanic induction (green curve), and 3 – the 2h after Tetanic induction (dark yellow curve). The fEPSP slope is the tangent of the tilt angle (α1, α2, and α3 for the 1st, 2d, and 3d state of slice throughout experiment, respectively) for the descending part of the postsynaptic response. The average values of the slopes during 20 min (1 measure every 30 s, total 40 sweeps) before the induction of LTP were taken as 100% (control).

3) Line 83: 188.9 ± 13.4%

Corrected.

4)Figure 2a needs in a more detailed description.

Corrected. The description was expanded.
